# *OsUGT88C3* Encodes a UDP-Glycosyltransferase Responsible for Biosynthesis of Malvidin 3-*O*-Galactoside in Rice

**DOI:** 10.3390/plants13050697

**Published:** 2024-02-29

**Authors:** Sihan Zhao, Shuying Fu, Zhenfeng Cao, Hao Liu, Sishu Huang, Chun Li, Zhonghui Zhang, Hongbo Yang, Shouchuang Wang, Jie Luo, Tuan Long

**Affiliations:** 1School of Breeding and Multiplication (Sanya Institute of Breeding and Multiplication), Hainan University, Sanya 572025, China; sihanzhao@hainanu.edu.cn (S.Z.); 21210901000006@hainanu.edu.cn (S.F.); 21220951310004@hainanu.edu.cn (Z.C.); 22220951310182@hainanu.edu.cn (H.L.); sishu.huang@hainanu.edu.cn (S.H.); chun.li@hainanu.edu.cn (C.L.); zhonghui.zhang@hainanu.edu.cn (Z.Z.); 22220951310107@hainanu.edu.cn (H.Y.); shouchuang.wang@hainanu.edu.cn (S.W.); 2School of Tropical Agriculture and Forestry, Hainan University, Haikou 570288, China; 3Hainan Yazhou Bay Seed Laboratory, Sanya 572025, China

**Keywords:** UDP-glycosyltransferase, *OsUGT88C3*, anthocyanin biosynthesis, malvidin, galactose

## Abstract

The diversity of anthocyanins is largely due to the action of glycosyltransferases, which add sugar moieties to anthocyanidins. Although a number of glycosyltransferases have been identified to glycosylate anthocyanidin in plants, the enzyme that catalyzes malvidin galactosylation remains unclear. In this study, we identified three rice varieties with different leaf color patterns, different anthocyanin accumulation patterns, and different expression patterns of anthocyanin biosynthesis genes (ABGs) to explore uridine diphosphate (UDP)-glycosyltransferases (UGTs) responsible for biosynthesis of galactosylated malvidin. Based on correlation analysis of transcriptome data, nine candidate UGT genes coexpressed with 12 ABGs were identified (r values range from 0.27 to 1.00). Further analysis showed that the expression levels of one candidate gene, *OsUGT88C3*, were highly correlated with the contents of malvidin 3-*O*-galactoside, and recombinant OsUGT88C3 catalyzed production of malvidin 3-*O*-galactoside using UDP-galactose and malvidin as substrates. OsUGT88C3 was closely related to UGTs with flavone and flavonol glycosylation activities in phylogeny. Its plant secondary product glycosyltransferase (PSPG) motif ended with glutamine. Haplotype analysis suggested that the malvidin galactosylation function of *OsUGT88C3* was conserved among most of the rice germplasms. *OsUGT88C3* was highly expressed in the leaf, pistil, and embryo, and its protein was located in the endoplasmic reticulum and nucleus. Our findings indicate that *OsUGT88C3* is responsible for the biosynthesis of malvidin 3-*O*-galactoside in rice and provide insight into the biosynthesis of anthocyanin in plants.

## 1. Introduction

Anthocyanins are a flavonoid group of phenylpropanoid compounds, which give various hues of blue, purple, and red to plant leaves, fruits, and flowers. They use their bright colors as visual signals to attract pollinators and seed dispersants [1,2], and play key roles in resisting ultraviolet radiation, low temperatures [3,4], drought stress [5], and microbial agents [6]. Moreover, anthocyanins promote human health by preventing neuronal disorders, inflammatory conditions, diabetes, obesity, cardiovascular diseases, and cancer [7].

Anthocyanins are constituted by an anthocyanidin aglycon bound to one or more sugar moieties. Modification of anthocyanins by glycosylation, methylation, and acylation adds versatility to their colors and stability [8]. Although common anthocyanidins in vascular plants are cyanidin, delphinidin, pelargonidin, malvidin, petunidin, and peonidin, most anthocyanins are derivatives of cyanidin, delphinidin, and pelargonidin [9]. In rice, the four most frequently reported anthocyanins are cyanidin-3-glucoside, peonidin-3-glucoside, cyanidin-3-rutinoside, and cyanidin-3-galactoside [10].

Anthocyanin biosynthesis genes (ABGs) and enzymes have been characterized in a range of plant species [11]. The first step in anthocyanin synthesis is catalysis by chalcone synthase (CHS), which leads to the formation of tetrahydroxychalcone, the basic skeleton of anthocyanins. Then, tetrahydroxychalcone is converted to anthocyanins through a series of reactions catalyzed by chalcone isomerase (CHI), flavanone 3-hydroxylase (F3H), flavonoid 3′-hydroxylase (F3′H), flavonoid 3′, 5′-hydrolase (F3′5′H), dihydroflavonol 4-reductase (DFR), anthocyanin synthase (ANS) and uridine diphosphate (UDP)-glycosyltransferases (UGTs) [8,10,11]. Finally, anthocyanins move from cytosol to the vacuole with the aid of transporters such as glutathione-S-transferase (GST) localized in the cytoplasm [10]. CHS, CHI, F3H, and F3′H are classified as early biosynthesis genes (EBGs), whereas the downstream genes of the pathway are classified as late biosynthesis genes (LBGs) [12].

The transcriptional activation of ABGs largely relies on MYB-type transcription factors (TFs) or MBW complexes comprising MYBs, basic helix–loop–helix (bHLH) TFs, and WD-repeat (WDR) protein [12]. For example, OsC1 (MYB), OsRb (bHLH), and OsTTG1/OsPAC1 form an MBW complex to regulate all ABGs in rice leaves, while OsP1 (MYB) can independently activate the transcription of EBGs [13,14].

UGTs constitute family 1 glycosyltransferases, which is the largest glycosyltransferase family in the plant kingdom. They catalyze the transfer of a sugar moiety from activated donors, UDP-sugars, to a wide range of acceptors, such as anthocyanidins, flavonoids, and phenylpropanoids [15]. Plant UGTs are characterized by the plant secondary product glycosyltransferase (PSPG) motif, a C-terminal consensus sequence comprising 44 amino acid residues, which represents the UDP-sugar binding site of the enzymes. The last residue in the PSPG motif was shown to determine the selectivity of sugar donors between UDP-glucose and UDP-galactose [16].

To date, a number of UGTs responsible for anthocyanidin glycosylation have been identified from several plant species. UGTs from kiwifruit, Aralia cordata, celery, and carrot can catalyze the galactosylation of cyanidin, peonidin, and pelargonidin using UDP-galactose as the sugar donor [17,18,19,20], and UGTs from rice can catalyze the glucosylation of cyanidin, malvidin, peonidin, procyanidin A1, and procyanidin B2. Nevertheless, the specific UGT responsible for the galactosylation of malvidin in plants has not been identified yet.

In the present study, three rice varieties with different leaf color patterns, different anthocyanin accumulation patterns, and different ABG expression patterns were identified to explore UGTs involved in malvidin galactosylation. Further analyses showed that *OsUGT88C3* encoded an enzyme responsible for the 3-*O-*galactosylation of malvidin. OsUGT88C3 was a unique glycosyltransferase with glutamine residue at the last position of its PSPG motif and was phylogenetically closer to flavonoid glycosyltransferases than anthocyanidin glycosyltransferases. Haplotype analysis suggested that the malvidin galactosylation function of *OsUGT88C3* was largely conserved among the rice germplasms. *OsUGT88C3* was highly expressed in the leaf, pistil, and embryo, and its protein was located in the endoplasmic reticulum and nucleus. These results indicate that OsUGT88C3 catalyzes the biosynthesis of malvidin 3-*O*-galactoside and broaden our understanding of anthocyanin biosynthesis.

## 2. Results

### 2.1. Identification of Rice Varieties with Different Leaf Color Patterns

In order to explore UGTs involved in malvidin galactosylation in rice, three rice varieties, HN, S2, and S4, with representative leaf color pattern at the trefoil stage were identified from a germplasm collection consisting of varieties with purple tissues. At the trefoil stage, the top first leaf (L1) of variety HN was green, the top second leaf (L2) was purple, and the purple color of the top third leaf (L3) was darker than L2. However, L1, L2, and L3 of variety S2 were all purple, while L1, L2, and L3 of variety S4 were all green (Figure 1a).

Next, the anthocyanin composition of the three varieties were examined. Figure 1b,c show that the accumulation patterns of the total anthocyanins in L1, L2, and L3 of the three varieties were different. We further conducted qualitative and quantitative analyses of the composition of anthocyanins with ultrahigh performance liquid chromatography/tandem mass spectrometry (UPLC-MS/MS). The results showed that free anthocyanidins, such as peonidin, pelargonidin, petunidin, and malvidin, were abundant in S4 leaves. Anthocyanins, such as cyanidin 3-*O*-glucoside, cyanidin 3-*O*-galactoside, cyanidin 3-*O*-rutinoside, and peonidin 3-*O*-glucoside, were abundant in S2 leaves. In HN leaves, free anthocyanidins and anthocyanins were both abundant (Figure 1d and Appendix A). These results indicate that different anthocyanin accumulation patterns lead to different leaf color patterns in rice.

### 2.2. Expression Levels of Anthocyanin Biosynthesis Genes Were Positively Correlated with Most Anthocyanin Contents

The expression levels of ABGs (Figure 2a) were examined by quantitative reverse transcription–polymerase chain reaction (qRT-PCR). As Figure 2b shows, the expression levels of all detected ABGs, except *OsF3H*, first decreased from L1 to L2, and then increased from L2 to L3 in variety HN. In contrast, the expression levels of all detected ABGs, except *OsF3′H*, first increased from L1 to L2, and then decreased from L2 to L3 in variety S4. In variety S2, the transcript levels of most ABGs, except *OsANS*, *OsP1*, *OsRb*, and *OsTTG1*, decreased progressively from L1 to L3. Correlation analysis showed that the expression levels of more than half of the ABGs were positively correlated with anthocyanin contents, except malvidin 3-*O*-galactoside (Figure 2c). These results implied the involvement of other UGTs in malvidin galactosylation.

### 2.3. Differentially Expressed Genes (DEGs) Were Enriched in Phenylpropanoid and Flavonoid Pathways

In order to explore DEGs among varieties HN, S2, and S4, we sequenced the transcriptomes of L1, L2, and L3 of variety HN, L1 and L3 of variety S2, and L2 of variety S4, and DEG analysis was performed. The results showed that 3381 up-regulated genes and 2372 down-regulated genes were detected between L1 and L2 in HN (Figure 3a and Appendix A); 4773 up-regulated and 3885 down-regulated DEGs were detected between L1 and L3 in HN (Figure 3b and Appendix A); 578 up-regulated and 548 down-regulated DEGs were detected between L2 and L3 in HN (Appendix A and Appendix A). In S2, 798 up-regulated and 458 down-regulated DEGs were detected between L1 and L3 (Figure 3c and Appendix A). KEGG enrichment analysis showed that DEGs between L1 and L2 (Figure 3d), and between L1 and L3 (Figure 3e) of HN were most enriched in the phenylpropanoid pathway. DEGs between L1 and L3 of S2 were enriched in both the phenylpropanoid and flavonoid pathways (Figure 3f). No enriched pathway was detected for DEGs between L2 and L3 of HN. These results indicate that DEGs were highly enriched in the phenylpropanoid and flavonoid pathways.

### 2.4. Exploration of UGT Genes Involved in Anthocyanin Biosynthesis

To further explore UGT genes responsible for anthocyanin biosynthesis, we calculated the correlation coefficients between the expression profiles of all the expressed genes and each ABG, namely, *OsCHS*, *OsCHI*, *OsF3H*, *OsF3′H*, *OsDFR*, *OsANS*, *Os3GT*, *OsGSTU*, *OsC1*, *OsP1*, *OsRb*, and *OsTTG1* (Appendix A and Appendix A). Nine genes annotated as glycosyltransferase in MSU Rice Genome Annotation Project Release 7 (http://rice.uga.edu/; accessed on 5 April 2022) were found to have correlation coefficients above 0.8 with most ABGs, such as *OsF3H*, *OsF3′H*, *OsDFR*, *OsANS*, *Os3GT*, *OsGSTU*, and *OsP1*, above 0.6 with *OsCHS* and *OsCHI*, and ranging from 0.27 to 0.70 with *OsC1* and *OsTTG1* (Table 1 and Appendix A). Seven of the nine genes had been identified as UGT genes, and six were found in DEGs (Table 1; see Section 2.3). As the protein encoded by *OsUGT706E2* (LOC_Os05g45200) and *OsUGT88C3* (LOC_Os01g53370) among the nine genes did not exhibit any glycosyltransferase activities towards flavones, flavonols, and flavanones [21], further research was conducted on these two genes.

### 2.5. OsUGT88C3 Is Responsible for Malvidin Galactosylation

Due to the high correlation between the expression level of *UGTs* responsible for anthocyanin biosynthesis and anthocyanin contents [20,25], qRT-PCR was first used to detect the expression levels of *OsUGT706E2* and *OsUGT88C3* in L1, L2, and L3 of HN, S2, and S4, and then, the correlation coefficients between the expression levels and the anthocyanin accumulation levels were calculated (Figure 2c). As shown in Figure 4a,b, the expression levels of *OsUGT88C3* were highly correlated with the contents of malvidin 3-*O*-galactoside (r = 0.77), and the expression levels of *OsUGT706E2* were positively correlated with the contents of delphinidin 3-*O*-galactoside (r = 0.26). These results strongly suggested that *OsUGT88C3* was involved in the galactosylation of malvidin.

To further investigate the function of *OsUGT88C3* and *OsUGT706E2*, their coding sequences (CDSs) were amplified from HN and S2, respectively, and cloned into pGEX-6p-1 expression vector with glutathione S-transferase fused to the N-terminus. In total, four recombinant proteins (OsUGT88C3-HN, OsUGT88C3-S2, OsUGT706E2-HN, and OsUGT706E2-S2) that were successfully expressed in *E*. *coil* were purified for enzymatic assays. Two sugar donors (UDP-glucose and UDP-galactose) and eleven typical flavonoid aglycones, namely, anthocyanidins (cyanidin, delphinidin, pelargonidin, malvidin, peonidin, petunidin), procyanidin (procyanidin C1), flavone (apigenin), flavonol (kaempferol), and dihydroflavonols (dihydrokaempferol and dihydroquercetin), were tested as potential substrates. The results showed that both OsUGT88C3-HN and OsUGT88C3-S2 could use UDP-galactose and malvidin as substrates, transferring galactose from UDP-galactose to malvidin to produce malvidin 3-*O*-galactoside (Figure 4c,d). The apparent Km and Kcat were calculated. The catalytic efficiency (kcat/Km) of OsUGT88C3-HN and OsUGT88C3-S2 were 0.29 min^−1^ μM^−1^ and 0.23 min^−1^ μM^−1^, respectively. In addition, the two OsUGT88C3 enzymes could also glucosylate malvidin in vitro (Appendix A). BdUGT88C8, the most related ortholog of OsUGT88C3, displayed activity of malvidin galactosylation and glucosylation as well (Appendix A). No catalytic activity was detected for either OsUGT706E2-HN or OsUGT706E2-S2, in accordance with previous reports [21].

### 2.6. OsUGT88C3 Is a Unique Enzyme Distinct from Other UDP-Glycosyltransferase

We subsequently performed a protein phylogeny reconstruction based on sequences of OsUGT88C3 and additional UGTs from family 1 glycosyltransferases reported in several plants (Figure 5a and Appendix A). According to the phylogenetic proximity, the phylogenetic tree could be divided into 16 clades, each containing UGTs with distinct substrate specificity and regiospecificity (Figure 5a).

OsUGT88C3 was clustered into the 7GT-3GT-II clade consisting of UGTs from the UGT88C subgroup [15]. In this clade, BdUGT88C8 was the only UGT with known function, which could attach glucoses to 7-OH of apigenin and 3-OH of kaempferol [21]. Additionally, in the 7GT-3GT-II clade, which was most phylogenetically close to the 7GT-3GT-I clade, OsUGT706D1, OsUGT706E1, BdUGT706E4, and BdUGT706F2 possess similar catalytic activities to BdUGT88C8, and OsUGT706D2 and OsUGT706F1 could catalyze the glucosylation of kaempferol and quercetin [21].

Moreover, OsUGT88C3 was phylogenetically far from other UGTs known to be responsible for anthocyanin biosynthesis, such as AcUFGT3a [20], VvF3GT1 [26], LcUFGT1 [27], Os3GT [28], AgUCGalT1 [19], DcUCGalT1 [18], AcGaT [17], and AtF3GlcT [29] in the 3GT clade; PfA5GlcT [30], VhA5GT [30], and AtF5GlcT [29] in the 5GT clade; OsGT1 [22] and OsGT5 [22] in the 7GT-5GT-3GT clade; OsGT4 [22] in the CGT clade; and OsGT2 [22], VpA3Glc2″GlcT [31], InA3Glc2″GlcT [32], AtA3Glc2″XylT [33], and AtF3Glc″GlcT [34] in the GGT clade. These results indicated that OsUGT88C3 is phylogenetically closer to flavonoid glycosyltransferase than anthocyanidin glycosyltransferase.

The last residues of the PSPG motif of OsUGT88C3 and 78 other UGTs reported before were analyzed (Figure 5b, Appendix A). The results showed that the last residue of the PSPG motif in most (4/6) of the galactosyltransferases was histidine, while the last residue of the PSPG motif in most (55/56) of the glucosyltransferases and most (3/4) of the glucuronosyltransferases was glutamine (Figure 5b and Appendix A). The PSPG motif of OsUGT88C3 ended with glutamine residue (Figure 5b).

### 2.7. Haplotype Analysis of OsUGT88C3

To investigate the natural variation in *OsUGT88C3*, we searched for variations within *OsUGT88C3* CDS in 4725 rice accessions on the database RiceVarMap [35]. The results showed that a total of 32 single nucleotide polymorphisms (SNPs) were found in the CDS of *OsUGT88C3*, of which 17 SNPs were synonymous variants and 15 SNPs were missense variants (Appendix A). PolyPhen-2 analysis showed that three SNPs, vg0130667310, vg0130667781, and vg0130667793, were possibly damaging variants (Appendix A), which are more likely to affect protein function [36]. Based on the three possibly damaging SNPs, haplotype analysis identified four haplotypes (I to IV; Figure 6). Haplotype I contained 4257 accessions, comprising various types of rice, including indica, aus, japonica, and intermediate rice. The genotypes of the three SNPs in haplotype I were all wildtype, and both varieties HN and S2 belonged to haplotype I. Haplotype II only included indica rice (154/154) and is the only haplotype with SNP vg0130667781 variant. Haplotype III was mainly composed of indica rice (218/224), with SNP vg0130667310 variant as the unique marker. Haplotype IV mainly consisted of japonica rice (89/90), characterized by the SNP vg0130667793 variant (Figure 6a,b). These results indicated that haplotype I was a functional haplotype, whereas haplotypes II, III, and IV were putatively nonfunctional haplotypes.

### 2.8. Subcellular Localization Analysis of OsUGT88C3

To examine the subcellular localization of OsUGT88C3, 35S:OsUGT88C3-green fluorescence protein (GFP) fusion vector was constructed and transiently transformed into *Nicotiana benthamiana* leaf epidermal cells. The results showed that green fluorescence was found to be located in the nucleus and endoplasmic reticulum, overlapping with red fluorescence emited by nucleus-located (Figure 7a) and endoplasmic reticulum-located (Figure 7b) mCherry markers [37]. These results indicated that OsUGT88C3 was located in the nucleus and endoplasmic reticulum.

### 2.9. OsUGT88C3 Is Highly Expressed in Leaves, Sheaths, Pistils, and Embryos

The expression levels of *OsUGT88C3* in roots, stems, stem nodes, leaves, sheaths, and panicles of HN at heading date were examined by qRT-PCR. *OsUGT88C3* was expressed in all tissues examined, but the expression levels were high in sheaths, low in roots, and moderate in other tissues (Appendix A). To examine the expression profile of *OsUGT88C3* further, chip data were extracted from the RiceXpro database (http://ricexpro.dna.affrc.go.jp/; accessed on 8 June 2023). The chip data showed that *OsUGT88C3* was highly expressed in leaves, sheaths, pistils, and embryos (Figure 7c). Taken together, these results indicated that *OsUGT88C3* was expressed at higher levels in leaves, sheaths, pistils, and embryos.

## 3. Discussion

### 3.1. Multiple UGTs Participate in the Synthesis of Anthocyanins in Rice Leaves

In this study, three varieties, HN, S2, and S4, were first identified according to their different leaf color patterns at the trefoil stage (Figure 1a). These three varieties exhibited different patterns of anthocyanin accumulation (Figure 1b–d) and different expression patterns of anthocyanin biosynthesis genes (Figure 2b). Correlation analysis showed that the expression levels of most of the tested ABGs, including *Os3GT*, were positively correlated with the contents of all detected cyanidin and peonidin glycoside (Figure 2c), in accordance with their functions in previous reports [28,38]. It was noted that malvidin 3-*O*-galactoside was not positively correlated with the expression levels of *Os3GT* and most of the other tested ABGs, implying that other UGTs were involved in malvidin glycosylation.

To explore other UGTs responsible for anthocyanin biosynthesis, the transcriptome of leaves of the three varieties were sequenced, and nine UGTs genes coexpressed with *Os3GT* and all the other ABGs were screened out (Table 1 and Appendix A). Among the nine genes, *OsGT1*, *OsGT2*, *OsGT4*, and *OsGT5* had been demonstrated to possess glycosyltransferase activities towards cyanidin and other anthocyanidin during germination [22]. *OsUGT90A1* had been shown to function in the protection of plasma membranes during chilling stress in rice [24]. *XYXT1* encoded a xylosyltransferase catalyzed the addition of 2-*O*-xylosyl side chains onto the xylan backbone [23]. *OsUGT707A3* exhibited 3′ specific glycosyltransferase activity towards flavonols, while *OsUGT706E2* and *OsUGT88C3* did not exhibit any glycosyltransferase activities towards flavones, flavonols, and flavanones [21]. Except *OsGT2*, *OsGT4*, and OsXYXT1, the other UGT genes were also DEGs (Table 1). The results suggest involvement of multiple UGTs in the synthesis of anthocyanin in rice leaves.

### 3.2. OsUGT88C3 Is Responsible for Malvidin Galactosylation

The previously identified UDP-galactose flavonoid 3-*O*-glycosyltranferases exhibited various substrate specificities. For example, AcUFGT3a recognized only cyanidin as substrates [20], while AcGaT, DcUCGalT1, and AgUCGalT1 could catalyze the galactosylation of both anthocyanidins (cyanidin, peonidin, and pelargonidin) and flavonols [17,18,19]. In our study, the recombinant OsUGT88C3 could only transfer galactose from UDP-galactose to 3-OH of malvidin (Figure 4c,d). In addition, the expression levels of *OsUGT88C3* was highly correlated with the accumulation levels of malvidin 3-*O*-galactoside (Figure 2c and Figure 4a,b). These results demonstrate that *OsUGT88C3* is responsible for malvidin galactosylation.

Anthocyanin was reported to be synthesized on the surface of the cell endoplasmic reticulum [10]. Anthocyanin glycosyltransferases were found to be localized to the endoplasmic reticulum and nucleus [20,39]. Our findings showed that OsUGT88C3 was located in the endoplasmic reticulum and nucleus (Figure 7a,b), consistent with the observations of other anthocyanin glycosyltransferases.

### 3.3. OsUGT88C3 Is a Unique Galactosyltransferase

In this study, amino acid sequences of 79 identified UGTs were collected to perform phylogenetic analysis (Appendix A). The 79 UGTs were clustered into 16 clades (Figure 5a). In the 79 UGTs, four galactosyltransferases (AcUFGT3a, AgUCGalT1, DcUCGalT1, and AcGaT) involved in anthocyanin biosynthesis all belonged to the 3GT clade [17,18,19,20]. Another two galactosyltransferases (SlGAME1 and StSGT1) involved in glycoalkaloid glycoside biosynthesis belonged to the MGT clade [40,41]. The glucosyltransferases involved in anthocyanin biosynthesis were clustered into the 3GT, 5GT, GGT, 7GT-5GT-3GT, and CGT clades (Figure 5a and Appendix A). These results are essentially consistent with previous findings [42,43]. Distinct from the above UGTs, OsUGT88C3 was clustered into the 7GT-3GT-II clade, which was closely related to UGTs responsible for flavone and flavonol glycosylation in phylogeny.

Multiple alignment analysis revealed that diverse amino acid residues existed at the last position of PSPG motifs in the 79 UGTs (Appendix A). The last residues of the PSPG motifs in nearly all (55/56) of the glucosyltransferases and most (3/4) of the glucuronosyltransferases were glutamine, while the majority (4/6) of the galactosyltransferases possessed PSPG motifs that ended with histidine. These results are in accordance with those of previous studies [16,44]. Additionally, AtF3AraT contained a PSPG motif that ended in histidine, but it used both UDP-glucose and UDP-arabinose as substrates [45]. SbUBGAT was a glucuronosyltransferase with a PSPG motif that ended in leucine [44]. SlGAME2 and AtA3Glc2″XylT were both xylosyltransferases, but their PSPG motif ended with asparagine and glutamine, respectively [33,46]. SlGAME1 and StSGT1 were galactosyltransferases, their PSPG motifs both ending with histidine [40,41]. OsUGT88C3 could use both UDP-galactose and UDP-glucose as substrates, and its PSPG motif ended with glutamine. The above results indicate that OsUGT88C3 is a unique galactosyltransferase distinct from other UGTs in rice.

### 3.4. The Catalytic Activity of OsUGT88C3 Is Conserved in Rice Germplasm

Based on the three possibly damaging SNPs, vg0130667310, vg0130667781, and vg0130667793, that were more likely to affect protein function within the *OsUGT88C3* CDS among 4725 rice accessions (Appendix A), the 4725 rice accessions were grouped into four haplotypes (I to IV). The genotypes of all the three SNPs in haplotype I were wildtype, suggesting that this haplotype was functional. Indeed, *OsUGT88C3* alleles in varieties HN and S2 were both haplotype I and displayed similar catalytic activity (Figure 4c,d). Considering that haplotype I contained 4257 accessions and comprised indica, aus, japonica, and intermediate types of rice, we suppose that the catalytic activity of OsUGT88C3 is conserved in rice germplasm.

Haplotypes II and III mainly consisted of indica rice, characterized by the SNP vg0130667781 and vg0130667310 variants, respectively. Haplotype IV was mainly composed of japonica rice, and was characterized by the SNP vg0130667793 variant. These results indicated that haplotypes II, III, and IV were putatively nonfunctional haplotypes, and that the three haplotypes were originated from haplotype I independently.

## 4. Materials and Methods

### 4.1. Plant Materials

Three rice (*Oryza sativa* L.) varieties, Heibaonuo (HN), Shenlvheishui2 (S2), and Shenlvheishui4 (S4), from the Metabolic Biology Laboratory of Hainan University were used in this study. Seeds of the three varieties were grown in growth chambers under controlled conditions (relative humidity 70%, average day/night temperature 25 °C, 12/12 h day/night cycle). The top first leaf (L1), the top second leaf (L2), and the top third leaf (L3) of HN, S2, and S4 varieties were harvested at the trefoil stage for further experiments.

### 4.2. Total Anthocyanin Content Measurement

Total anthocyanin content measurement was performed as described by Liu et al. [47], with minor modifications. Briefly, the total anthocyanins were extracted from 0.1 g powdered leaf sample at 60 °C for 2 h in the dark using 10 mL extraction buffer (prepared from 95% ethanol with a volume ratio of 85:15 and 1.5 mol/L hydrogen chloride). The extracts were then centrifuged at 12,000× *g* for 5 min and filtered through a 0.45 μM filter. OD values of the supernatant were determined by an ultraviolet spectrophotometer (Thermo Fisher Scientific NanoDrop One, Waltham, MA, USA) at a length wave of 535 nm. One OD is equivalent to one anthocyanin unit. The content of anthocyanins was expressed in units per gram.

### 4.3. Metabolic Sample Preparation and Profiling

In brief, rice samples were first freeze-dried and then ground into powder using Ball Mill BM500 (Anton Paar, Graz, Austria) for 1 min at 30 Hz. Next, 100 mg of the powder was extracted at 4 °C for 12 h with 1 mL pre-cooled methanol in the dark, followed by centrifugation for 10 min at 12,000× *g* and 4 °C. The supernatants were filtrated (SCAA-104, 0.22 mm pore size; ANPEL Shanghai, China) and then analyzed using an ultrahigh-performance liquid chromatography/tandem mass spectroscopy (UPLC-MS/MS)-based targeted method described previously by Zhang et al. [22] and Yang et al. [48].

### 4.4. RNA Extraction and Quantitative Reverse Transcription Polymerase Chain Reaction

Total RNA was isolated from rice tissues using TRIzol reagent (Invitrogen, Carlsbad, CA, USA), according to the manufacturer’s instructions. RNA concentration and integrity were determined by spectrophotometer (Thermo Fisher Scientific, NanoDropOne, USA) and 1.0% agarose gel electrophoresis, respectively. Approximately 1 μg of total RNA was used to synthesize first-strand cDNA with ToloScript All-in-One RT EasyMix for qPCR (TOLOBIO, Shanghai, China).

Real-time quantitative polymerase chain reaction was performed on a QuantStudio7 Flex Real-Time PCR System instrument (Thermo Fisher Scientific, USA) using 2 × Q3 SYBR qPCR Master Mix (TOLOBIO, Shanghai, China). The amplification program consisted of 40 cycles of 95 °C for 30 s, followed by 95 °C for 10 s and 60 °C for 30 s. The amplification was normalized by the 2^−ΔΔCT^ method [49], using the endogenous reference gene GAPDH as an internal standard. The values reported represent the average of three biological replicates.

### 4.5. RNA Sequencing and Data Analysis

RNA samples were sent to Novogene Corporation (Beijing, China) for RNA sequencing. Reference genome and gene information files were downloaded from MSU Rice Genome Annotation Project Release 7 (http://rice.plantbiology.msu.edu; accessed on 5 April 2022). An Illumina Hi Step 4000 instrument generated raw reads, and the sequencing read length was double-ended 2 × 150 base pairs (bp) (PE150) and processed with fastq to filter out adapters and low-quality sequences. HISAT2 v.2.1.0 (http://ccb.jhu.edu/software/hisat2/index.shtml; accessed on 3 May 2022) was used for mapping clean reads to the reference genome. The gene expression level was quantified using the feature counts based on the expected number of transcripts per kilobase of the exon model per million mapped reads (TPM) method. Differential expression analysis was performed using the DESeq2 R package in Bioconductor version 1.30.0 with adjusted *p*-values based on TPM values. The differential expression levels were determined using the Benjamini–Hochberg FDR multiple testing correction with an adjusted *p*-value < 0.05 and |log2fold change| > 1.2, according to the literature [50].

FeatureCounts v1.5.0-p3 was used to count the reads numbers mapped to each gene. Then, fragments per kb of transcript per million fragments mapped (FPKMs) of each gene were calculated based on the length of the gene and reads count mapped to this gene. Genes with FPKM values greater than 0.3 in at least one sample were considered to be expressed, and were used to calculated correlation coefficients matrices with the expression profiles of each ABG using R/CORRGRAM. A heatmap of resulting correlation coefficients was illustrated using R/PHEATMAP.

### 4.6. KEGG Analysis

Kyoto Encyclopedia of Genes and Genomes (KEGG) analysis was performed to identify differentially expressed genes (DEGs) that were significantly enriched in metabolic pathways at Bonferroni-corrected *p*-values ≤ 0.05 compared with the whole-transcriptome background. KEGG pathway analysis was carried out using clusterProfiler R package in Bioconductor version 3.18.0.

### 4.7. Recombinant Protein Expression and In Vitro Enzyme Assay

The coding sequences (CDSs) of *OsUGT88C3* and *OsUGT706E2* were amplified from varieties HN and S2 (Appendix A), respectively, and cloned into pGEX-6p-1 expression vector (Novagen) with glutathione S-transferase tag. The pGEX-6p-1 expression vector containing *OsUGT88C8* CDS was kindly provided by Dr. Peng [21]. The expression and purification of recombinant proteins followed the description by Peng et al. (2017). Target proteins were confirmed by SDS-PAGE.

The in vitro enzyme assay for the recombinant proteins was performed as follows: a total volume of 100 μL reaction containing 200 μM anthocyanidin substrate, 1.5 mM UDP galactose, 5 mM Mgcl_2_, 500 ng of total purified protein, and Tris-HCl buffer (100 mM, pH 7.4) was first incubated at 30 °C for 20 min, and then, it was stopped by the addition of 300 μL of ice-cold methanol. The resulting mixture was filtered through a 0.2 μM filter (microporous) and analyzed by LC-MS. The LC-MS conditions followed the description by Peng et al. [21].

### 4.8. Phylogenetic Analysis and Multiple Sequence Alignment

The amino acid sequences for the genes of interest were downloaded from NCBI (https://www.ncbi.nlm.nih.gov/; accessed on 10 June 2023), and then aligned using the MUSCLE algorithm in MEGA 11 software (version MEGA 11 (64-bit); http://megasoftware.net/; accessed on 17 June 2023). The alignments were then used to construct phylogenetic trees using the neighbor-joining method based on the Poisson model. Bootstrap values were calculated using 1000 replicates.

### 4.9. Haplotype Analysis

The CDS of *OsUGT88C3* (LOC_Os01g53370) from varieties HN and S2 were directly sequenced by PCR products, and then subjected to genotyping. The SNPs and InDels within *OsUGT88C3* CDS in 4725 rice accessions were downloaded from the RiceVarMap [35] website (http://ricevarmap.ncpgr.cn/; accessed on 22 July 2023). All the SNPs obtained were used for haplotype analysis by DNASP v.5 (http://www.ub.edu/dnasp/; accessed on 22 July 2023), and the haplotype network was drawn by a haplotype viewer (http://www.cibiv.at/~greg/haploviewer; accessed on 29 July 2023). SnpEff version 5.1d and CooVar version 1.0 softwares were used to evaluate the impact of variations. PolyPhen-2 version 2.0 was employed to quantitatively evaluate missense mutation effect [36].

### 4.10. Subcellular Localization

The CDSs of *OsUGT88C3* were amplified (Appendix A) and cloned into the pJC032 vector, in frame with *enhanced green fluorescent protein* (*eGFP*). The pJC032 empty vector was used as the positive control. A nucleus-located marker and an endoplasmic reticulum (ER)-located marker [51] each labeled with mCherry were used to locate the fluorescent proteins in the nucleus and ER, respectively. The above vectors were transformed into *Agrobacterium tumefaciens* strain GV3101 competent cells, and were injected into 6-week-old *Nicotiana benthamiana* leaves in different combinations. After incubation for 48–72 h at 24 °C under a 16 h/8 h (day/night) photoperiod, the fluorescence was observed with a laser scanning confocal microscope LSM 980 (Carl Zeiss AG, Oberkochen, Germany).

## 5. Conclusions

In summary, three rice varieties with different leaf color patterns, anthocyanin accumulation patterns, and ABG expression patterns were first identified. Using correlation analyses, nine candidate UGT genes highly coexpressed with the ABGs were screened out from the varieties. Among the candidate genes, *OsUGT88C3* was highly correlated with the contents of malvidin 3-*O*-galactoside. Enzymatic analysis demonstrated that recombinant OsUGT88C3 catalyzed galactosylation of malvidin on its 3-OH using UDP-galactose and malvidin as substrates. OsUGT88C3 was phylogenetically far from other anthocyanidin glycosyltransferases but close to flavone and flavonol glycosyltransferases, and its PSPG motif ended with glutamine. Haplotype analysis showed that most rice germplasms retained the functional allele of *OsUGT88C3*. *OsUGT88C3* was highly expressed in leaf, sheaths, pistil and embryo, and its protein was located in the endoplasmic reticulum and nucleus. Our research indicates that OsUGT88C3 is a unique and conserved enzyme responsible for malvidin galactosylation in rice.

## Figures and Tables

**Figure 1 plants-13-00697-f001:**
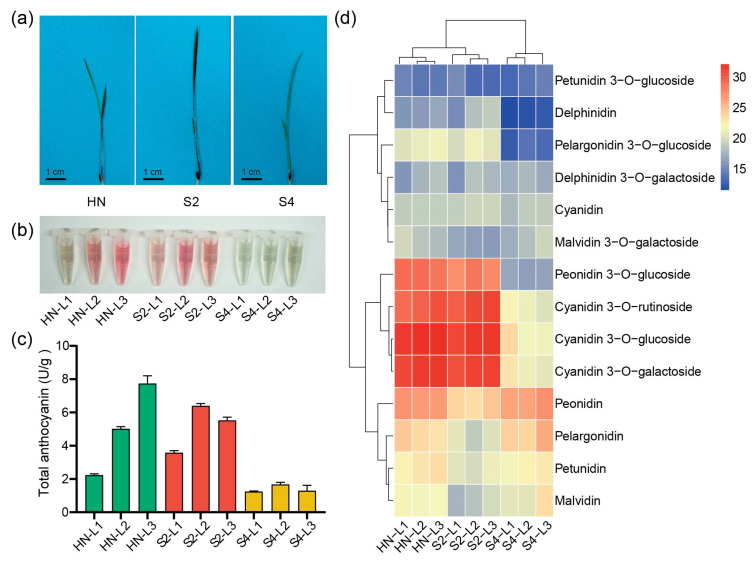
Phenotypes and anthocyanin accumulation patterns of rice varieties, HN, S2, S4. (**a**) Phenotypes of HN, S2, S4 seedling at trefoil stage, scale bar = 1 cm. (**b**) The anthocyanin extracts of the top first leaf (L1), the top second leaf (L2), and the top third leaf (L3) of HN, S2, S4 at trefoil stage, respectively. (**c**) Total anthocyanin contents of L1, L2 and L3 of HN (green columns), S2 (red columns), and S4 (yellow columns) at trefoil stage. Data represent the mean ± SD of three independent experiments. (**d**) Hierarchically clustered heatmap of contents of 14 types of anthocyanins detected in L1, L2, and L3 of HN, S2, S4. Red and blue indicate high and low abundance, respectively.

**Figure 2 plants-13-00697-f002:**
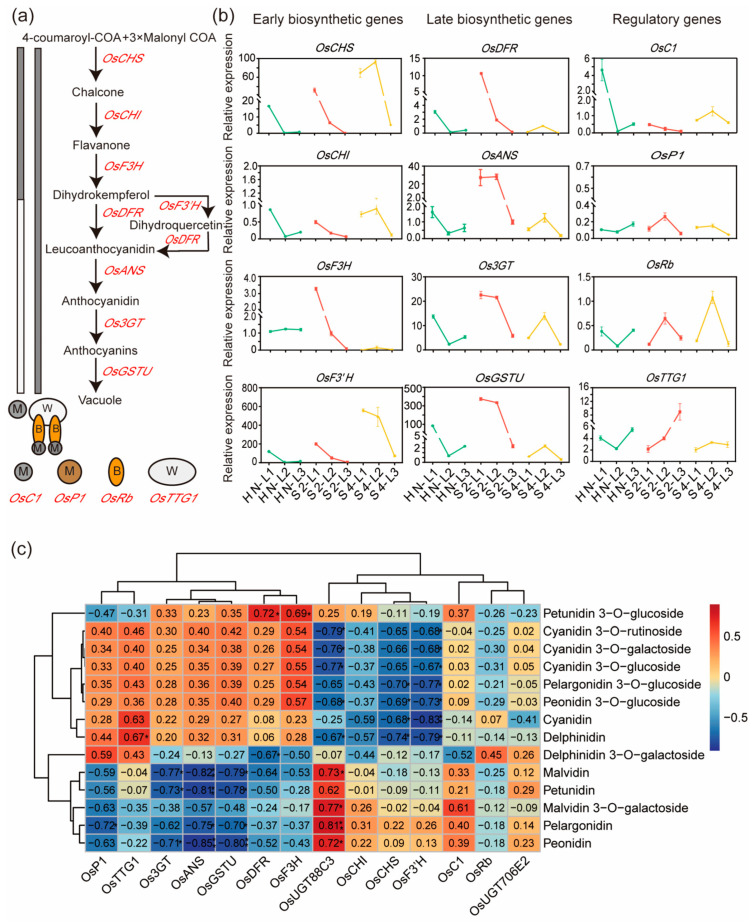
The expression levels of anthocyanin biosynthesis genes and their correlation with anthocyanin contents in rice leaves. (**a**) The biosynthesis pathway of anthocyanin in rice. (**b**) The expression levels of early biosynthesis genes including OsCHS, OsCHI, OsF3H, and OsF3′H, late biosynthesis genes including OsDFR, OsANS, Os3GT, and OsGSTU, and regulatory genes including OsC1, OsP1, OsRb and OsTTG1 in the top first leaf (L1), the top second leaf (L2), and the top third leaf (L3) of rice varieties HN (green lines and dots), S2 (red lines and dots), S4 (yellow lines and dots) at trefoil stage are present. The data are presented as the mean ± SE of three independent experiments. (**c**) Correlation coefficients between expression levels of anthocyanin biosynthesis genes and anthocyanin contents in leaves (* *p* < 0.05, ** *p* < 0.01).

**Figure 3 plants-13-00697-f003:**
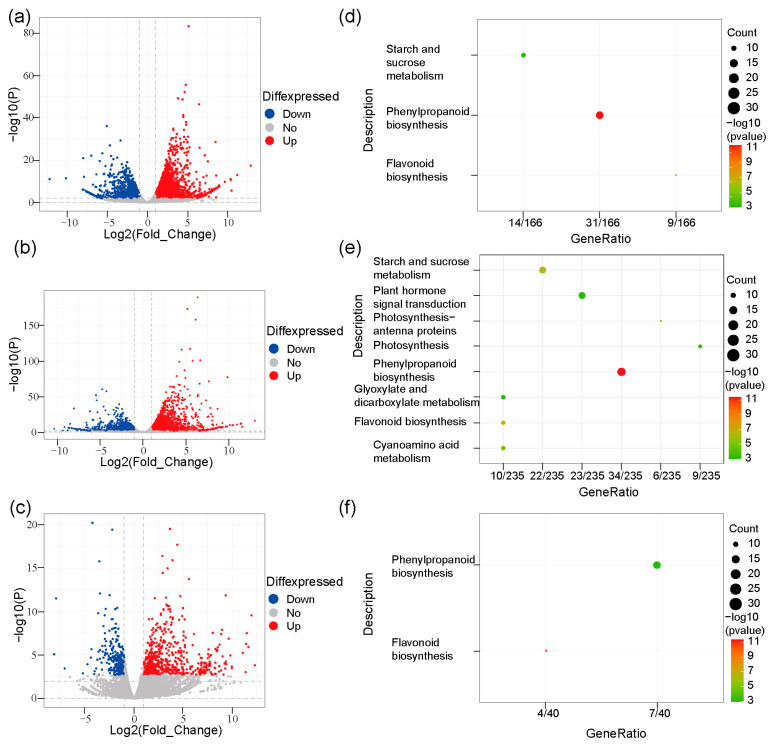
Comparative transcriptome analysis for rice varieties, HN, S2, S4. The volcano plots (**a**–**c**) and KEGG enrichment analysis (**d**–**f**) of differentially expressed genes between the top first leaf (L1), and the top second leaf (L2) in HN (**a**,**d**), L1 and the top third leaf (L3) in HN (**b**,**e**), L1 and L3 in S2 (**c**,**f**).

**Figure 4 plants-13-00697-f004:**
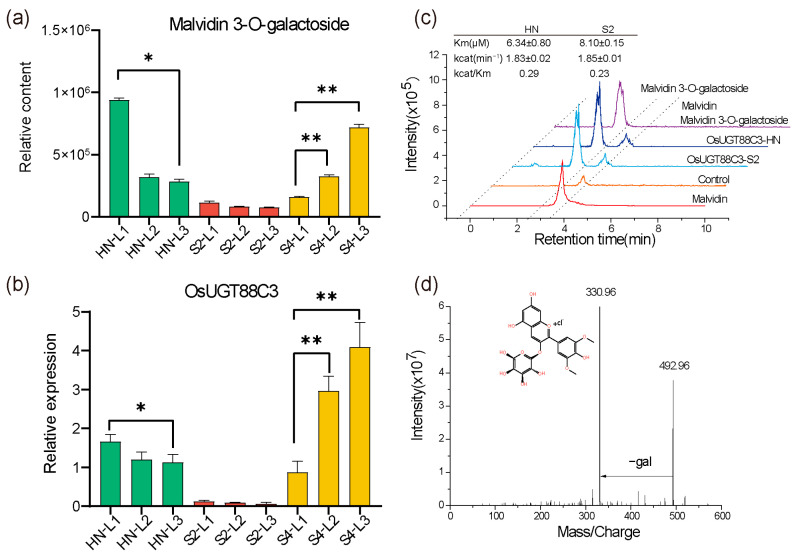
Functional analysis of *OsUGT88C3*. Malvidin 3-*O*-galactoside contents (**a**) are highly correlated with *OsUGT88C3* expression levels (**b**) in the top first leaf (L1), the top second leaf (L2), and the top third leaf (L3) of rice varieties HN, S2, S4 at trefoil stage. The data are presented as mean ± SD, *n* = 3. * *p* < 0.05, ** *p* < 0.01, Student’s *t*-tests. (**c**) HPLC chromatograms of the in vitro reaction of OsUGT88C3 with UDP-galactose and malvidin. Kinetic parameters of OsUGT88C3 from varieties HN (OsUGT88C3-HN) and S2 (OsUGT88C3-S2) are shown. Data are mean values ± SD, calculated from two biological experiments. (**d**) The MS spectrum and chemical structure of the product in the reaction of OsUGT88C3 with UDP-galactose and malvidin.

**Figure 5 plants-13-00697-f005:**
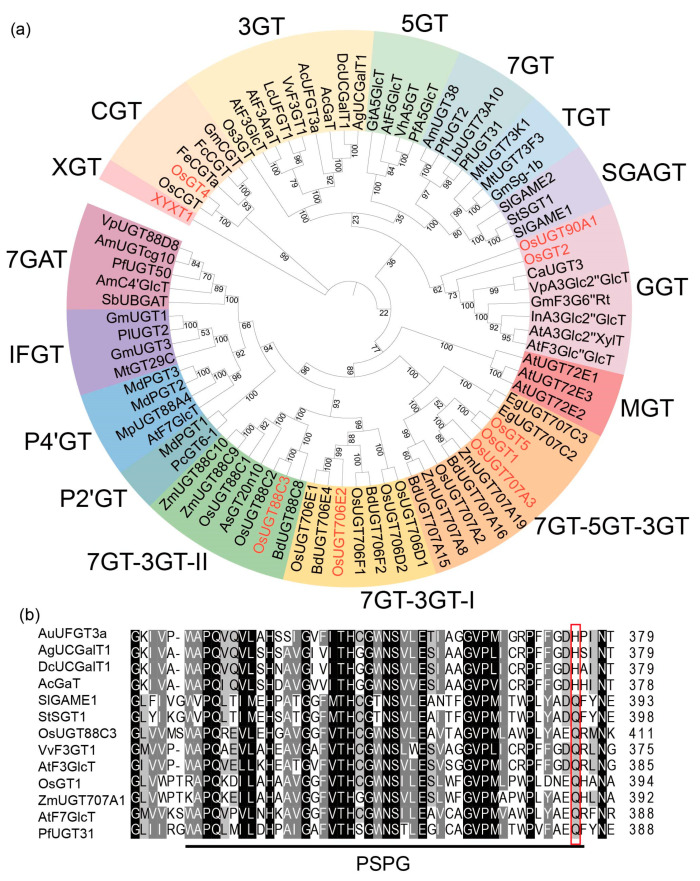
Phylogenetic analysis and amino acid sequence alignment of OsUGT88C3 and UDP-glycosyltransferases from other plants. (**a**) Phylogenetic tree of selected plant UGTs and OsUGT88C3. Sixteen functional clades are shaded in different colors. The nine candidate UGTs found in this study are represented in red font. XGT, Xylan glycosyltransferase; CGT, flavonoid C-glycosyltransferases; 3GT, flavonoid 3-*O*-glycosyltransferases; 5GT, flavonoid 5-*O*-glycosyltransferases; TGT, triterpenoid glycosyltransferase; SGAGT, steroidal glycoalkaloid glycosyltransferase; GGT, flavonoid glycoside glycosyltransferases; MGT, monolignol glycosyltransferase; 7GT-5GT-3GT, UGTs with at least one of 7-*O*-, 5-*O*-, and 3-*O*-glycosyltransferase activity; 7GT-3GT-I, group I UGTs with at least one of 7-*O*- and 5-*O*-glycosyltransferase activity; 7GT-3GT-II, group II UGTs with at least one of 7-*O*- and 5-*O*-glycosyltransferase activity; P2′GT, phloretin 2′-*O*-glycosyltransferase; P4′GT, phloretin 4′-*O*-glycosyltransferase; IFGT, isoflavone glycosyltransferase; 7GAT, 7-*O*-glucuronosyltransferases. The abbreviated species names and all sequences are provided in Appendix A. (**b**) Multiple sequence alignment of OsUGT88C3 with other UDP-glycosyltransferases. The black line under the sequences indicates the putative conserved secondary plant glycosyltransferase (PSPG) motifs. The red box indicates the last residue of the PSPG motif.

**Figure 6 plants-13-00697-f006:**
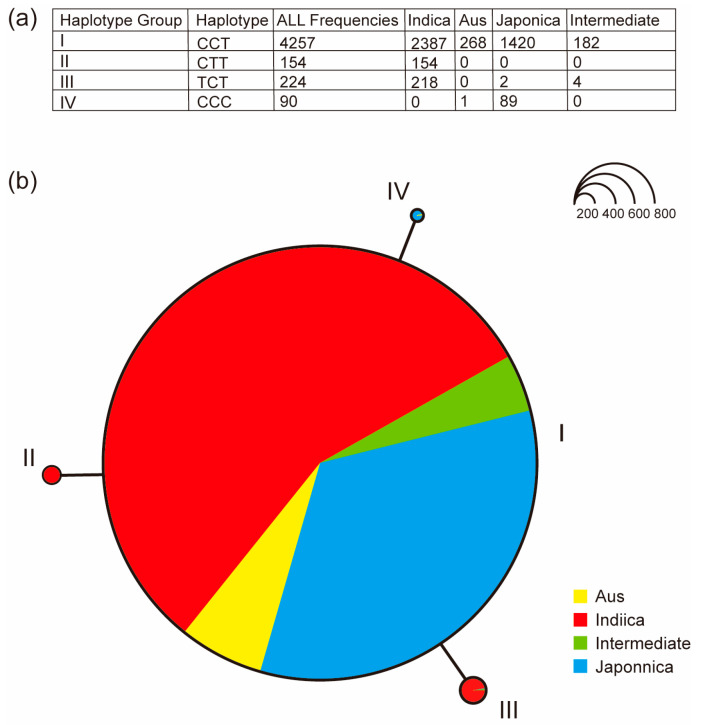
Haplotype analysis of *OsUGT88C3*. (**a**) Sequence polymorphism of different haplotypes of *OsUGT88C3*. (**b**) Haplotype network of *OsUGT88C3*.

**Figure 7 plants-13-00697-f007:**
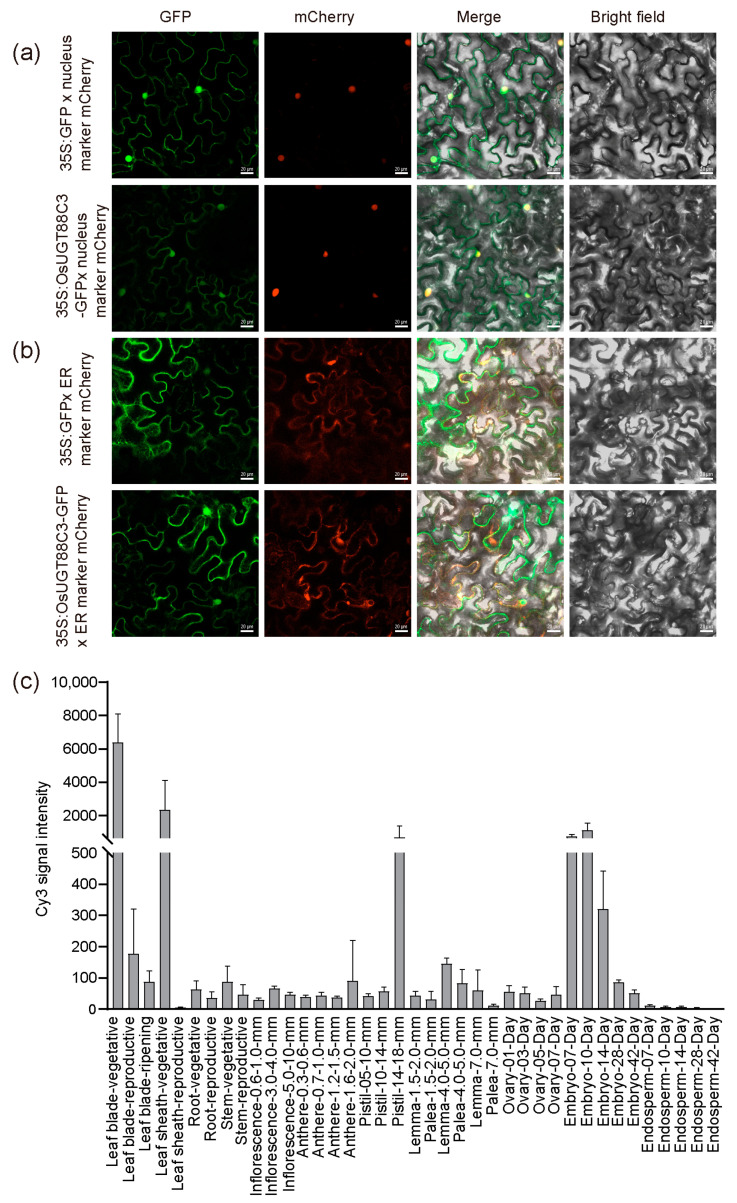
Subcellular localization and expression pattern of *OsUGT88C3*. (**a**) Subcellular localization of OsUGT88C3 in *Nicotiana benthamiana* treated with 35S:GFP or 35S:OsUGT88C3-GFP and a nuclei-located marker labeled with mCherry. (**b**) Subcellular localization of OsUGT88C3 in *Nicotiana benthamiana* treated with 35S-GFP or 35S:OsUGT88C3-GFP and an ER-located marker labeled with mCherry. Scale bars: 20 μm. (**c**) Expression pattern of *OsUGT88C3*. Leaf, root, and stem were collected at 12 o’clock. Error bars show the mean ± SD.

**Table 1 plants-13-00697-t001:** Nine candidate UGT genes involved in anthocyanin biosynthesis.

Locus	Gene Symbol	Tissue ^a^	Annotation ^b^	Function ^c^	Reference
LOC_Os04g24110	*OsUGT707A3*	HN-L3/HN-L1, S2-L3/S2-L1	UDP-glucoronosyl and UDP-glucosyl transferase domain containing protein, expressed	Glucosylates 3-OH of kaempferol	[21]
LOC_Os05g45200	*OsGT4*, *OsCGT*	Not detected	UDP-glucoronosyl and UDP-glucosyl transferase domain containing protein, expressed	Glucosylates cyanidin, malvidin, peonidin, and procyanidin A1	[22]
LOC_Os07g32620	*OsUGT88C3*	HN-L3/HN-L1, S2-L3/S2-L1	Anthocyanidin 5,3-*O*-glucosyltransferase, putative, expressed	No activity on flavones, flavonols, and flavanones	[21]
LOC_Os02g37690	*OsGT1*	HN-L3/HN-L1, S2-L3/S2-L1	UDP-glucoronosyl and UDP-glucosyl transferase domain containing protein, expressed	Catalyzes cyanidin as a substrate to form cyanidin 3-*O*-glucoside	[22]
LOC_Os06g49300	*OsXYXT1*	Not detected	Glycosyltransferase protein, putative, expressed	Catalyzes the addition of 2-*O*-xylosyl side chains onto the xylan backbone	[23]
LOC_Os07g32630	*OsGT2*	Not detected	UDP-glucoronosyl and UDP-glucosyl transferase, putative, expressed	Glucosylates cyanidin, malvidin, and procyanidin B2	[22]
LOC_Os01g53370	*OsGT5*	HN-L3/HN-L1, S2-L3/S2-L1	Anthocyanidin 5,3-*O*-glucosyltransferase, putative, expressed	Glucosylates cyanidin, peonidin, and procyanidin A1	[22]
LOC_Os06g18010	*OsUGT706E2*	HN-L3/HN-L2, S2-L3/S2-L1	Anthocyanidin 5,3-*O*-glucosyltransferase, putative, expressed	No activity on flavones, flavonols, and flavanones	[21]
LOC_Os07g32020	*OsUGT90A1*	HN-L3/HN-L1, HN-L3/HN-L2, S2-L3/S2-L1S2-3/S2-1	Anthocyanin 3-*O*-beta-glucosyltransferase, putative, expressed	Helps protect plasma membranes during chilling stress in rice	[24]

^a^ Differential expression tissues among the top first leaf (L1), the top second leaf (L2), and the top third leaf (L3) of rice varieties HN, S2, S4 at trefoil stage. ^b^ Annotation in MSU Rice Genome Annotation Project Release 7 (http://rice.uga.edu/; accessed on 5 April 2022). ^c^ Function reported in the reference.

## Data Availability

All data and materials are available on request.

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
