# Peer review of "OsUGT88C3* Encodes a UDP-Glycosyltransferase Responsible for Biosynthesis of Malvidin 3-*O*-Galactoside in Rice"

_plants, 2024, doi:10.3390/plants13050697_

Round 1
Reviewer 1 Report
Comments and Suggestions for Authors
In this manuscript, the authors identified three rice cultivars with different leaf color patterns, different anthocyanin content, and different gene expression. Based on the analysis, nine candidate UGT genes co-expressed with ABG were identified. It should be noted that the authors found that the expression levels of one candidate gene, OsUGT88C3, are highly correlated with the content of malvidin 3-O-galactoside.
The manuscript is well written, the results are clearly presented. In my opinion, the conclusions should be presented a little better, they are too short. I suggest that this manuscript be accepted for publication, with minor corrections.
Comments on the Quality of English LanguageIn this manuscript, the authors identified three rice cultivars with different leaf color patterns, different anthocyanin content, and different gene expression. Based on the analysis, nine candidate UGT genes co-expressed with ABG were identified. It should be noted that the authors found that the expression levels of one candidate gene, OsUGT88C3, are highly correlated with the content of malvidin 3-O-galactoside.
The manuscript is well written, the results are clearly presented. In my opinion, the conclusions should be presented a little better, they are too short. I suggest that this manuscript be accepted for publication, with minor corrections.
Author Response
Response to Reviewer 1 Comments
Thank you very much for taking the time to review this manuscript. Please find the Point-by-point response to comments and suggestions for authors below and the corresponding revisions and corrections in track changes in the re-submitted files.
Comments and Suggestions for Authors
In this manuscript, the authors identified three rice cultivars with different leaf color patterns, different anthocyanin content, and different gene expression. Based on the analysis, nine candidate UGT genes co-expressed with ABG were identified. It should be noted that the authors found that the expression levels of one candidate gene, OsUGT88C3, are highly correlated with the content of malvidin 3-O-galactoside.
The manuscript is well written, the results are clearly presented. In my opinion, the conclusions should be presented a little better, they are too short. I suggest that this manuscript be accepted for publication, with minor corrections.
Comments on the Quality of English Language
In this manuscript, the authors identified three rice cultivars with different leaf color patterns, different anthocyanin content, and different gene expression. Based on the analysis, nine candidate UGT genes co-expressed with ABG were identified. It should be noted that the authors found that the expression levels of one candidate gene, OsUGT88C3, are highly correlated with the content of malvidin 3-O-galactoside.
The manuscript is well written, the results are clearly presented. In my opinion, the conclusions should be presented a little better, they are too short. I suggest that this manuscript be accepted for publication, with minor corrections.
Response: Thanks for your valuable comments. We have accepted your feedback and rewrite the conclusions accordingly. The conclusions after rewriting is as follows: In summary, three rice varieties with different leaf color patterns, anthocyanin accumulation patterns, and ABG expression patterns were first identified. Using correlation analyses, nine candidate UGT genes highly coexpressed with the ABGs were screen out from the varieties. Among the candidate genes, OsUGT88C3 was highly correlated with the contents of malvidin 3-O-galactoside. Enzymatic analysis demonstrated that recombinant OsUGT88C3 catalyzed galactosylation of malvidin on its 3-OH using UDP-galactose and malvidin as substrates. OsUGT88C3 was phylogenetically far from other anthocyanidin glycosyltransferases but close to flavone and flavonol glycosyltransferases, and its PSPG motif was ended with glutamine. Haplotype analysis showed that most rice germplasm retained the functional allele of OsUGT88C3. OsUGT88C3 was highly expressed in leaf, sheaths, pistil and embryo, and its protein was located in endoplasmic reticulum and nucleus. Our research indicates that OsUGT88C3 is a unique and conserved enzyme responsible for malvidin galactosylation in rice.

Reviewer 2 Report
Comments and Suggestions for Authors
The research article "OsUGT88C3 encodes a UDP-glycosyltransferase responsible for biosynthesis of malvidin 3-O-galactoside in rice" by Sihan Zhao et al., focuses on galactosylated malvidin biosynthesis genes.
The article is well written, I found no issues with any part of it. The only point in materials and methods is that the use of pH-differential method for determination of antocyanins concentration (Lee et al.: JOURNAL OF AOAC INTERNATIONAL Vol. 88, No. 5, 2005) would be preferable since it would give an absolute quantitative concentration and is more robust than method used by the Authors.
I think this article is of interest to the readers and should be published in current form, only some proofreading is needed to correct small typos like:
Line 222: "was phylogenetic far from other" -> was phylogenetically far from other
Line 234: "The PSPG motif of was OsUGT88C3 was ended" -> The PSPG motif of OsUGT88C3 was ended.
Author Response
Response to Reviewer 2 Comments
Thank you very much for taking the time to review this manuscript. Please find the Point-by-point response to comments and suggestions for authors below and the corresponding revisions and corrections in track changes in the re-submitted files.
Comments and Suggestions for Authors
The research article "OsUGT88C3 encodes a UDP-glycosyltransferase responsible for biosynthesis of malvidin 3-O-galactoside in rice" by Sihan Zhao et al., focuses on galactosylated malvidin biosynthesis genes.
The article is well written, I found no issues with any part of it. The only point in materials and methods is that the use of pH-differential method for determination of anthocyanins concentration (Lee et al.: JOURNAL OF AOAC INTERNATIONAL Vol. 88, No. 5, 2005) would be preferable since it would give an absolute quantitative concentration and is more robust than method used by the Authors.
Response: Thanks a lot for your valuable comments. We studied the pH-differential method in the reference (Lee et al.: JOURNAL OF AOAC INTERNATIONAL Vol. 88, No. 5, 2005), and found that this method has been widely used in many studies (Zhu, F.; Cai, Y.-Z.; Bao, J.; Corke, H. Effect of γ-irradiation on phenolic compounds in rice grain. Food Chem. 2010, 120, 74-77; Zheng, J.; Wu, H.; Zhu, H.; Huang, C.; Liu, C.; Chang, Y.; Kong, Z.; Zhou, Z.; Wang, G.; Lin, Y.; et al. Determining factors, regulation system, and domestication of anthocyanin biosynthesis in rice leaves. New Phytol. 2019, 223, 705-721). Although the method we used in this study is simpler (Liu, M.; Wang, Z.; Gu, Y. Caryopsis Development and Anthocyanidin Accumulation of Colored Rice. Chin. J. Rice Sci./Zhongguo Shuidao Kexue 2011, 25, 392-398.), we agree with your opinion that the pH-differential method will give an absolute quantitative concentration and be more robust. We will adopt the pH-differential method in further research.
I think this article is of interest to the readers and should be published in current form, only some proofreading is needed to correct small typos like:
Line 222: "was phylogenetic far from other" -> was phylogenetically far from other
Response: Thank you for your comments. We have made correction accordingly.
Line 234: "The PSPG motif of was OsUGT88C3 was ended" -> The PSPG motif of OsUGT88C3 was ended.
Response: Thank you for the comments. We have made correction according to your comments.

Reviewer 3 Report
Comments and Suggestions for Authors
Minor revision

NA
Author Response
Response to Reviewer 3 Comments
Thank you very much for taking the time to review this manuscript. Please find the Point-by-point response to comments and suggestions for authors below and the corresponding revisions and corrections in track changes in the re-submitted files.
- Abstract, lines 19. Clarify the co-expression between UGT gene and ABGs gene.
Response: Thank you very much for your comment. The co-expression between UGT gene and ABGs gene have been clarified in Results 2.4, lines 153 - lines 160. The details are as follows: to further explore UGT genes responsible for anthocyanin biosynthesis, we calculated correlation coefficients between expression profiles of all the expressed genes and each ABGs, including OsCHS, OsCHI, OsF3H, OsF3¢H, OsDFR, OsANS, Os3GT, OsGSTU, OsC1, OsP1, OsRb and OsTTG1 (Figure S2 and Table S6). Nine genes annotated as glycosyltransferase in MSU Rice Genome Annotation Project Release 7 (http://rice.uga.edu/) were found to have correlation coefficients above 0.8 with most ABGs, such as, OsF3H, OsF3¢H, OsDFR, OsANS, Os3GT, OsGSTU, and OsP1, above 0.6 with OsCHS and OsCHI, range from 0.27 to 0.70 with OsC1 and OsTTG1 (Table 1 and Table S6).
Due to the length limitation of the abstract and the results above, we clarify the co-expression between UGT gene and ABGs gene in the abstract as follows: Based on correlation analysis of transcriptome data, nine candidate UGT genes coexpressed with 12 ABGs were identified (r values range from 0.27 to 1.00). Further analysis showed that the expression levels of one candidate gene, OsUGT88C3, were highly correlated with the contents of malvidin 3-O-galactoside, and recombinant OsUGT88C3 catalyzed production of malvidin 3-O-galactoside using UDP-galactose and malvidin as substrates.
We have made corresponding revisions in the abstract.
- Line 39. Space needed after “cancer”.
Response: Thanks for your comment. We have added a space after “cancer”.
- Line 55. Space needed after “menbrane”.
Response: Thanks for your detailed comment. We have added a space after “menbrane”.
- Line 58. Space needed.
Response: Thanks for your suggestions. We have added a space after “(LBGs)”.
- Line 59-61. More clarification is needed.
Response: We accept the comment and clarify our opinions in a following new paragraph. The context of the new paragraph is as follows: The transcriptional activation of ABGs largely relies on MYB-type transcription factors (TFs) or MBW complexes comprising MYBs, basic helix-loop-helix (bHLH) TFs, and WD-repeat (WDR) protein [12]. For example, OsC1 (MYB), OsRb (bHLH), and OsTTG1/OsPAC1 form a MBW complex to regulate all ABGs in rice leaves, while OsP1 (MYB) can independently activate the transcription of EBGs [13,14].
- Line 94-95. Modify the sentence according to figure 1a.
Response: Thanks for the comment. We change the sentence “However, all of the L1, L2 and L3 were purple and green in varieties S2 and S4, respectively” to “However, L1, L2, and L3 of variety S2 were all purple, while L1, L2, and L3 of variety S4 are all green”
- Line 121. Figure doesn’t represent the statement. In figure 2b at OsF3H, an increasement can
be noticed from L1 to L2 and then decreased from L2 to L3 in S4 variety.
Response: We accept the comment and correct the statement as “In contrast, the expression levels of all detected ABGs, except OsF3′H, first increased from L1 to L2, and then, decreased from L2 to L3 in variety S4.”
- Line 178. Remove space.
Response: We are sorry for the mistake and remove the space after “0.77”.
- Line 225. Reference needed.
Response: Thanks for your comment. We have provided the reference in the corresponding positions in the text.
- Line 226. Reference needed.
Response: Thanks for your comment. We have provided the reference in the corresponding positions in the text.
- Line 273. Figure 6b is misleading. Color representation and the illustration according to the 4 different types of rice should be appropriate.
Response: Thanks for your suggestion. We have modified the color representation and the illustration in Figure 6b.
- Line 277. Scientific name should be in Italic form.
Response: We change "Nicotiana benthamiana" to italics.
- Line 284. Convert the formation of the Scientific name to Italic form.
Response: We change "Nicotiana benthamiana" to italics.
- Line 285. Convert the formation of the Scientific name to Italic form.
Response: We change "Nicotiana benthamiana" to italics.
- Line 319. Grammatical error.
Response: Thanks for the comment. We have corrected “suggests” to “suggest”.
- Line 388. Grammatical error.
Response: Thanks for the comment. We have corrected “was” to “were”.
- Line 409. Misspelled word.
Response: We are sorry for the mistake. We have corrected “Syatem” to “System”.
- Line 435. Grammatical error.
Response: Thanks for your comment. We have corrected “was” to “were”.
- Line 468. Change the lower case to upper case.
Response: Thanks for your comment. We have corrected “snpEff” to “SnpEff”.
- Line 561. Don’t find the paper in Google Scholar
Response: Thanks for the comment. We are sorry that we are unable to visit Google Scholar in China. However, the paper (Liu, Y.; Zhou, B.; Qi, Y.; Liu, C.; Liu, Z.; Ren, X. Biochemical and functional characterization of AcUFGT3a, a galactosyltransferase 561 involved in anthocyanin biosynthesis in the red-fleshed kiwifruit (Actinidia chinensis). Physiol. Plant. 2018, 162, 409-426.) can be found in NCBI Pubmed library (https://pubmed.ncbi.nlm.nih.gov/29057484/; Figure 1) and WILEY Online Library (https://onlinelibrary.wiley.com/doi/full/10.1111/ppl.12655; Figure 2).
Figure 1
Figure 2
